# *Thymus atlanticus*: A Source of Nutrients with Numerous Health Benefits and Important Therapeutic Potential for Age-Related Diseases

**DOI:** 10.3390/nu15184077

**Published:** 2023-09-21

**Authors:** Adil El Midaoui, Farid Khallouki, Réjean Couture, Florina Moldovan, Mahmoud Ali Ismael, Brice Ongali, Marie Yvonne Akoume, Chakib Alem, Ali Ait Boughrous, Wafa Zennouhi, Mhammed Chaoui Roqai, Lhoussain Hajji, Imen Ghzaiel, Anne Vejux, Gérard Lizard

**Affiliations:** 1Department of Pharmacology and Physiology, Faculty of Medicine, University of Montreal, Montreal, QC H3C 3J7, Canada; rejean.couture@umontreal.ca (R.C.); ongali.brice@gmail.com (B.O.); 2Department of Biology, Faculty of Sciences and Techniques, Errachidia, Moulay Ismail University of Meknes, Meknes 50050, Morocco; farid_khallouki@yahoo.fr (F.K.); boughrous@gmail.com (A.A.B.); w.zennouhi@edu.umi.ac.ma (W.Z.); 3Research Center of CHU Sainte Justine, Faculty of Dentistry, Université de Montréal, Montreal, QC H3T 1J4, Canada; florina.moldovan@umontreal.ca (F.M.); my.akoume@gmail.com (M.Y.A.); 4Northgate Centre 2074-9499 137 Ave NW, Edmonton, AB T5E 5R8, Canada; 5Research Team in Biochemistry and Natural Resources, Faculty of Sciences and Techniques, Moulay Ismail University of Meknes, Meknes 20250, Morocco; alem04@yahoo.fr; 6Ecole des Hautes Etudes de Biotechnologie et de Santé (EHEB), 183 Bd de la Résistance, Casablanca 20250, Morocco; direction@eheb.ma; 7Laboratory of Bioactives and Environmental Health, Faculty of Sciences, Moulay Ismail University, Meknes 50050, Morocco; hajjime@hotmail.fr; 8Laboratory “Biochemistry of the Peroxisome, Inflammation and Lipid Metabolism”, Bio-peroxIL/EA7270, Université de Bourgogne/Inserm, 21000 Dijon, France; imenghzaiel93@gmail.com (I.G.); anne.vejux@u-bourgogne.fr (A.V.)

**Keywords:** *Thymus atlanticus*, botany, cultivation, phytochemistry, biological activities, age-related diseases

## Abstract

*Thymus atlanticus* (Lamiaceae) is a plant endemic to the Mediterranean basin that is found in significant quantities in the arid regions of Morocco. *Thymus atlanticus* is used in traditional medicine to treat infectious and non-infectious diseases. It is also used for the isolation of essential oils and for the seasoning of many dishes in the Mediterranean diet. The major constituents of *Thymus atlanticus* are saponins, flavonoids, tannins, alkaloids, various simple and hydroxycinnamic phenolic compounds, and terpene compounds. Several of these compounds act on signaling pathways of oxidative stress, inflammation, and blood sugar, which are parameters often dysregulated during aging. Due to its physiochemical characteristics and biological activities, *Thymus atlanticus* could be used for the prevention and/or treatment of age-related diseases. These different aspects are treated in the present review, and we focused on phytochemistry and major age-related diseases: dyslipidemia, cardiovascular diseases, and type 2 diabetes.

## 1. History, Geographical Distribution, and Use of ***Thymus atlanticus*** (Lamiaceae) in Traditional Medicine

Nowadays, Thymus species are versatile herbs that are cultivated and encountered worldwide. The genus Thymus contains about 350 species of aromatic perennial herbaceous plants. Thymus possibly derives from the Greek word “thyo”, which means to perfume, cleanse, or fumigate, or refers to the Greek word “thymon”, which signifies bravery or a cure [1]. The ancient Egyptians used it for the purpose of embalming. For the Greeks, thyme was a symbol of grace and elegance, and they also used it as incense for ritual purposes. Thyme originates from the Mediterranean basin and came to Western Europe with the soldiers of the Roman legions. In ancient Rome, it was recommended for lung diseases and to prevent poisoning, along with its properties of enhancing bravery and strength as well as restoring vigor [2]. The sprigs were thought to provide protection against the plague and were burned indoors to purify the air. Thyme can complement the treatment of infectious respiratory diseases since it is recommended as an expectorant and as an antimicrobial agent, as well as to eliminate digestive disorders in Persian traditional medicine. A number of Thymus species are used as digestive, antispasmodic, carminative, antitussive, and expectorant agents [3]. Thyme contributes to its traditional uses in culinary preparations as a flavorful herb and in cosmetic products. It is worth noting that there are various cultivars and hybrids of thyme available, each with its own unique characteristics and flavors. Some popular cultivars include lemon thyme (*Thymus citriodorus*), caraway thyme (*Thymus herba-barona*), and creeping thyme (*Thymus serpyllum*). Common thyme (*Thymus vulgaris*) has been considered the most usable. There is also a Spanish variety (*Thymus zygis*), also known as white thyme. Furthermore, diluted thyme oil rubbed into the scalp removes dandruff, stimulates circulation, improves hair condition, and accelerates its growth. It also has a relaxing effect. Thyme is also an ornamental plant with significant decorative values. It is also used as an additive in preparations used in the tanning process of leather.

The Moroccan endemic species *Thymus atlanticus* (Ball) Roussine is locally known by its vernacular name: in Amazigh as Azukni, Tazuknite, and in Arabic as Ziitra. In infusion or decoction, its leafy stems are used against headaches, influenza, fever, and chills, as well as against digestive disorders and menstrual pain in women. Its fumigation is used to treat respiratory ailments. It is important to note that while thyme has a long history of traditional use, scientific research is ongoing to fully understand its potential health benefits and any associated risks.

The genus Thymus belongs to the Lamiaceae family, which is one of the largest families in the plant kingdom and one of the eight most important genera in terms of the number of species included [4]. According to Tzima et al. [5], the genus Thymus belongs to the 250 most diverse genera and 7200 species [5]. As stated by Dob et al. [6], there are nearly 350 species of thyme distributed between Europe, including Greece, France, Turkey, Italy, and Spain, and they are relatively well acclimatized in other European countries, Asia, and South-West Arabia, as well as in the Mediterranean basin, where they are mostly concentrated [3]. Indeed, the plant is native to the Mediterranean basin and is cultivated due to its adaptability in the aforementioned regions beyond its native range. Additionally, different species and varieties of thyme may thrive in other arid and semi-arid regions in different climates and environmental conditions, resulting in a wide distribution across the globe, including Oceania and certain African countries outside the Mediterranean region.

In Morocco, the Lamiaceae family has more than 225 species, characterized by great wealth and biodiversity. The genus Thymus is represented by about twenty species, of which 12 are endemic [7]. Moroccan thyme species are found in the plains, in the mountains, in rockeries, scrubland, lawns, or scrub [8,9]. The endemic species *Thymus atlanticus* (Ball) Roussine [10,11], which is very abundant locally, may be threatened by excessive harvesting. It is the smallest of the Moroccan thymes and very polymorphic, depending on the altitude and the ecology. It is encountered in arid, semi-arid, and sub-humid bioclimates with hot, temperate, and cool variants in the infra-Mediterranean, thermo-Mediterranean, and meso-Mediterranean vegetation stages. In Morocco, this species is distributed in the Anti Atlas, High Atlas, and Middle Atlas.

## 2. Botany and Cultivation

Thymus grows well in arid conditions with lots of sunlight. They thrive in well-drained calcareous soil and do not require fertile soil, with an optimum soil pH between 5.5 and 6.5. The optimum germination temperature is around 20 °C [2]. Thymus species are spontaneous, perennial, evergreen, aromatic, and very hardy herbaceous sub-woody shrubs. They are generally low-growing, reaching a height of 10 to 40 cm. The shrubs are characterized by tight, slender, erect, hairy twigs covered with opposite, short-stalked, ovate-oblong grayish-green, oblong-lanceolate to linear stems. The leaves are 4–10 mm long and elliptical to oblong in shape, glabrous but slightly ciliated at the base, slightly curled at the edges, and have short erect to prostrate quadrangular and glandular trichomes [12]. The stamens protruding from the upper lip of the corolla have divergent filaments. Reproduction takes place sexually (seed) and asexually (stump bursting, cuttings, and layering) [11]. The tiny, capitate zygomorphic flowers are axillary and grouped at the tip of the branches, forming a kind of terminal node, and their color varies from white to purple, passing through pink, attached to the top of the shoots. They are very polymorphic, mainly calcifuge, with glandular, glabrous, or slightly hairy calyx [10]. The fruit is tetraquenium and brown in color. Thyme flowers from May to the end of September. Thyme has a lifespan ranging from 4 to 7 years. Moreover, other Moroccan species exist, such as *Thymus atlanticus* (Ball) Roussine, which is the smallest of the Moroccan thymes. Indeed, it is a calcifuge, hemicryptophyte, perennial herbaceous plant with overwintering buds at ground level, specific to the Oro-Mediterranean floor of the Maghreb, with very small white or pale pink flowers, in low carpet, prostrate, very polymorphic, especially calcifuge, from 1700 to 3400 m (Figure 1) [10].

The taxonomic information on *Thymus atlanticus* is provided in Figure 2.

Thyme species are still very poorly studied botanically, phytochemically, and pharmacologically. Therefore, the objective of the present review was to update knowledge of *Thymus atlanticus* phytochemistry and its effects, as well as its compounds, on major age-related diseases: dyslipidemia, cardiovascular diseases, and type 2 diabetes.

## 3. Phytochemistry

Most of the thymus species remain chemically unexplored. Some data in the chemistry of the Thymus genus involves the presence of secondary metabolites that contribute to their distinctive aromas and potential medicinal bioactivities. Thymus genus bio-synthesizes important compounds of commercial interest; among them, essential oils stand out. These are a complex mixture of volatile compounds that are gaining increasing interest and are responsible for their characteristic scents and flavors. In particular, Thymus vulgaris and *Thymus zygis* are of great economic importance due to their high amounts of thymol and their concomitant isomer, carvacrol. Consequently, these species are mentioned in numerous international monographs [1].

Apart from thymol and carvacrol, the main constituents of thyme essential oils include p-cymene, linalool, terpinene, pinene, myrcene, and limonene. The occurrence of different chemotypes has been described in several species of Thymus plants. The Moroccan thyme essential oils are characterized by the chemotypes camphor, carvacrol, borneol, thymol, and geranyl acetate [13,14]. Although thyme grows industrially in several countries with a wide variety of climatic conditions, the best yields are obtained in countries with a Mediterranean climate, i.e., a climate with constant sunshine, such as Spain, France, Morocco, and Algeria. The average yield of thyme essential oil varies from 1 to 3%. Authors such as Golparvar and Bahari [15], Golmakani and Rezaei [16], and Salehi et al. [17,18] suggest that harvesting thyme at the beginning of flowering in the plant life cycle may help to obtain the highest yield of essential oils. The composition of the bioactive compounds and their amount, as well as their chemotypes, are influenced by the part of the plant used, the vegetative stage of the plant, the environmental conditions, and the time of harvest [19]. This variability has also been potentially explained by geographical regions, ecologic factors, local abiotic factors (topography, moisture, temperature, pedologic and edaphic factors), selective biotic factors, including associated fauna and flora [20], as well as genetic factors [21], and infraspecific variability [22,23]; Amarti et al. [24]; Stahl-Biskup and Saez [4]. This genus is subject to extensive intraspecific polymorphism, as exemplified by the common thyme (*Thymus vulgaris*), the best-known of which has indeed six chemotypes [25]. The two best-known chemotypes are thymol thyme and linalool thyme. The other four chemotypes of Thymus vulgaris are thujanol thyme, geraniol thyme, carvacrol thyme, and p-cymene thyme. Therefore, the specific composition and content of these essential oil components can vary between different thymus species and within cultivars of the same species. In most of the reported studies, there is a linear relationship between the concentration of thymol and carvacrol, i.e., when thymol is high in an essential oil, the amount of carvacrol is low, and vice versa. The chemical composition of essential oils of *Thymus atlanticus* in different phenological stages showed carvacrol as the main constituent, except for essential oil from leaves in the post-flowering stage, where the main constituent was thymol [26,27,28]. The γ-terpinene and p-cymene components are also phenolic precursors. Thymol is a naturally occurring monoterpene phenolic that derives from p-cymene and is an isomer of carvacrol. γ-terpinene and borneol are isomeric hydrocarbons that exhibit differences in the location of their carbon-carbon double bonds in their chemical skeletons [29].

Up to date, the chemistry of *Thymus atlanticus* metabolites is very sparse, with the exception of its essential oils and a few reported polyphenols. The essential oil of *Thymus atlanticus* was characterized by 21 volatile constituents, the main ones being carvacrol (47.1%), p-cymene (7.5%), and α-pinene (3.7%). Figure 3 depicts the structures of different systems in thyme essential oils found in the Moroccan *Thymus atlanticus* variety on a non-polar TG-5MS column [30].

The other key phytoconstituents of the thyme genus include saponins, flavonoids, tannins, alkaloids, and various simple and hydroxycinnamic phenolic compounds [31]. More particularly, hydroxycinnamic acids include ferulic acid, caffeic acid, and rosmarinic acid. The latter is the ester of caffeic acid with 3,4-dihydroxyphenyl lactic acid and is commonly known as a caffeic acid dimeric form, the most abundant phenolic in Thymus species. As indicated in Figure 4, there are other polyphenolic compounds including, apigenin, luteolin, naringenin, thymonin, dihydroquercetin, eriodictyol, quercetin, naringenin, apigenin, catechin, rutin, apigetrin, caffeic acid, p-coumaric acid, and chlorogenic acid [32]. The flavonoid content of thyme species plays an important role in the taxonomy and distinction between the different species [20]. The biosynthesis of terpenic compounds, which are derived from classical mevalonate pathways, also includes triterpenes such as ursolic acid and oleanolic acid (Figure 4).

Thyme also contains vitamins, especially A and C. Its leaves are an excellent source of potassium, calcium, iron, manganese, magnesium, and selenium [33]. It is important to note that the concentrations and proportions of such metabolites can also vary depending on the variety of thyme, growing conditions, and processing methods. 

Together, *Thymus atlanticus* may be a rich source of biologically active components. Indeed, Kouya et al. [34] have described non-volatile metabolites in the aqueous extract of *Thymus atlanticus,* notably rosmarinic acid, rutin, hyperoside, quercetin, apigetrin, and caffeic acid. High-performance liquid chromatography (HPLC) was and is still the main technique applied in the analysis of plant phenolics since it allows rapid, cost-effective qualitative and quantitative screening. This technique has also been combined with spectral information gathered by the photodiode array detector and, more particularly, by mass spectrometry analysis, which has been mainly carried out in the negative ion mode due to its high sensitivity in detecting distinct classes of phenolic compounds. By using these techniques, more studies will help to isolate new active compounds, allowing the identification of their exact mechanisms of action.

## 4. Benefits of ***Thymus atlanticus*** on Human Health

There are currently a lot of arguments about the benefits of *Thymus atlanticus* on human health. The data obtained with essential oils is convincing for a lot of microorganisms. The therapeutic effects of *Thymus atlanticus* on age-related diseases could be mediated, at least in part, by major plant compounds such as rosmarinic acid, apigenin, and quercetin. In this context, the individual effects of these different molecules on different in vitro and in vivo models are well documented. However, no data is currently available on the cocktail effects of these molecules, which could permit the development of new treatments and the use of *Thymus atlanticus* extracts in optimal conditions for various diseases.

### 4.1. Antimicrobial Activities of Thymus atlanticus

Previous investigations have reported that Moroccan endemic thymes exhibit important biological properties, including antifungal, antimicrobial, and antiviral activities [35,36,37]. Indeed, extracts from *T. riatarum, T. maroccanus,* and *T. broussonetii,* such as essential oils, were found to exert high antimicrobial efficacy in combination with standard antibiotics; the effect on Gram-positive bacteria was more important than on Gram-negative bacteria [35,36,37,38]. The results indicate that the oils had a high inhibitory activity against *E. coli*, *Salmonella* sp., Ent. Cloacae, *K. pneumonia*, *V. cholerae*, *B. subtilus*, *B. cereus*, *M. luteus,* and *S. aureus*, except for *Pseudomonas aeruginosa* [35]. In vitro studies have shown that *Thymus atlanticus* essential oils possess strong antimicrobial activity, notably against Escherichia coli [39]. Moreover, studies have shown that *Thymus atlanticus* essential oils exert synergic antimicrobial activity with ciprofloxacin against *Pseudomonas aeruginosa* and *Bacillus subtilis*. The maximum synergistic effect was observed for *Klebsiella pneumonia* [30]. Moreover, Thymus flavonoids such as apigenin and luteolin were reported to exert partially positive effects against antibiotic resistance [40]. Therefore, one may suggest that Thymus atlanticus utilization alone or in combination with antibiotics could represent an alternative method to prevent and control the emergence of resistant microbial strains.

### 4.2. Benefits of Thymus atlanticus on Age-Related Diseases

#### 4.2.1. Dyslipidemia

Studies have shown that polyphenol-rich extracts of *Thymus atlanticus* ameliorated Triton-induced acute hyperlipidemia in rats, hamsters, and mice [41,42,43]. *Thymus atlanticus* polyphenol-rich extracts were also found to be efficacious in reducing hyperlipidemia, notably total cholesterol and LDL-cholesterol levels, in chronically high-fat diet-treated hamsters [44]. The anti-hyperlipidemic effect of the *Thymus atlanticus* polyphenol-rich extract was partially explained by its ability to inhibit cholesterol biosynthesis, as reflected by the reduction in the hepatic expression of HMG-CoA reductase observed in chronically high-fat diet-treated hamsters [45]. The same authors reported that a polyphenol-rich extract of *Thymus atlanticus* significantly decreased total cholesterol content (*p* < 0.05) and HMG-CoA reductase expression (*p* < 0.05) but did not affect fecal cholesterol, bile acid contents, or CYP7A1 and ABCG5/G8 expression (*p* > 0.05) in high-fat diet-treated hamsters [45]. Interestingly, apigenin, a *Thymus atlanticus* polyphenolic compound, was found to attenuate the high cholesterol-feeding-induced hypercholesterolemia in hamsters [46].

#### 4.2.2. Atherosclerosis

Atherosclerosis is a major histological modification of the arteries that promotes the development of cardiovascular diseases and is also involved in vascular dementia [47,48]. The identification of phytomolecules that allow for the prevention of atherosclerosis is therefore of great interest. Previous studies have shown that the polyphenol extract of *Thymus atlanticus* inhibited inflammation caused by local application of croton oil or xylene and injection of carrageenan or arachidonic acid in animal models [34,43]. Moreover, the polyphenol extract of *Thymus atlanticus* was found to inhibit the production of monocyte chemoattractant protein 1 (MCP-1) by macrophage cultures activated by lipopolysaccharides in a dose-dependent manner [34]. Interestingly, paraoxonase-1, a well-known molecule playing a central role in protecting arteries against atherosclerosis [49], was found to be ameliorated in association with decreased lipid peroxidation in the polyphenol extract of *Thymus atlanticus* combined with fat-diet-treated hamsters [44]. These effects could be mediated by rosmarinic acid and apigenin. Previous studies have demonstrated that rosmarinic acid inhibited the progression of atherosclerotic plaque in association with a decrease in lipids, TNF-α, and IL-1β levels in atherosclerotic ApoE−/− mice [50]. Rosmarinic acid exerts its beneficial effects on atherosclerosis by inhibiting inflammasome activation in endothelial cells as well as promoting macrophage cholesterol efflux [51,52]. Moreover, naringenin, another compound of Thymus, was reported to modulate the biomarkers of vascular dysfunction and to protect the endothelium against unresolved inflammation, oxidative stress, and atherosclerosis [53], as well as to enhance aortic protein expression and activity of SIRT1 in aged Apoe−/− mice [54]. In addition, apigenin was found to prevent the development of atherosclerosis by inhibiting LOX-1 gene expression and increasing the Bcl-2/Bax ratio in hyperlipidemia rats [55,56].

#### 4.2.3. Myocardial Infarction

To date, no research has been undertaken to examine the possible effect of *Thymus atlanticus* on myocardial infarction. However, rosmarinic acid, its principal compound, was found to exert potent cardioprotective effects against acute myocardial infarction induced by isoproterenol in a rat model [57,58]. The mechanism underlying these effects was thought to be mediated by its ability to enhance expression of plasma antioxidant enzymes as well as genes involved in Ca^2+^ homeostasis of the sarcoplasmic reticulum Ca^2+^ ATPase (SERCA2) and ryanodine receptor (RyR2) [58] and by suppressing the elevation of malondialdehyde levels both in the serum and the myocardium [57]. More recently, Quan et al. [59] have suggested that rosmarinic acid may exert a cardioprotective effect against myocardial ischemia/reperfusion injury via suppression of the NF-κB (nuclear factor-kappa B) inflammatory signaling pathway and ROS (reactive oxygen species) production in mice. Furthermore, studies have shown that rosmarinic acid exerts protective effects against cardiac dysfunction and fibrosis following myocardial infraction through decreasing angiotensin-converting enzyme (ACE) expression and increasing ACE2 expression via the AT1R/p38 MAPK pathway [60]. Moreover, quercetin, a *Thymus atlanticus* flavonoid compound, has been associated with improved cardiovascular health [61]. In randomized controlled intervention trials evaluating different types of micronutrients, quercetin showed moderate- to high-quality evidence for decreasing CVD risk factors [62]. In addition, rutin’s bioglycosylated form was found to improve cellular therapy for myocardial infarction by enhancing the engraftment and differentiation potential of mesenchymal stem cells through suppressing ROS production [63].

#### 4.2.4. Hypertension

Studies have demonstrated that rosmarinic acid treatment improved blood pressure in fructose-fed [64] and angiotensin II-induced hypertensive rats [65]. The mechanisms underlying the anti-hypertensive effects of rosmarinic acid were found to be mediated through its antioxidant properties [64,66], angiotensin-converting enzyme ACE inhibition [60,67], and vasodilation [64]. Indeed, rosmarinic acid was found to reduce blood pressure via suppression in the expression of NADPH oxidase (nicotinamide adenine dinucleotide phosphate oxidase) p22phox subunit [64], decrease in ACE activity [60,67], endothelin-1 (ET-1) levels, and angiotensin type 1 receptor (AT1R) expression, as well as via an increase in nitric oxide (NO) production [64]. Moreover, other investigations have shown that rosmarinic acid results in a decrease in blood pressure in association with a reduction in serum corticosterone levels and proinflammatory cytokine production in chronically hypertensive angiotensin II-treated rats [68]. Although rosmarinic acid seems to exert beneficial effects on experimental hypertension, human studies should be undertaken in order to confirm if such positive effects could be achievable at clinically attainable concentrations per dose. Interestingly, studies have shown that quercetin reduces blood pressure in spontaneously hypertensive rats [69]. Moreover, quercetin was found to attenuate the rise in blood pressure and aortic thickness of the abdominal aorta in association with reversed expression of differentially expressed transcripts and signaling pathways in the abdominal aorta of Ang II-infused C57BL/6 mice [70].

#### 4.2.5. Stroke

Previous studies have shown that rosmarinic acid alleviates neurological deficits and reduces cerebral infarct volumes in association with the attenuation of decreased superoxide dismutase, catalase activities, and GSH (reduced glutathione) levels in rats suffering from ischemic stroke and post-stroke depression [71]. Animal studies have suggested that rosmarinic acid exerts a neuroprotective effect against ischemic stroke through a mechanism involving the modulation of the Nrf2/HO-1 pathway [72]. Rosmarinic acid has been primarily applied as a key ingredient in Chinese herbal medicine for clinical ischemic stroke treatment. In one randomized controlled trial, 500 mg of rosmarinic acid appeared to be safe and tolerable in healthy individuals [73]. Interestingly, experimental studies have shown that quercetin alleviates cerebral infusion/reperfusion-induced neurological deficits, brain infarction, blood-brain barrier disruption, oxidative stress, TNF-α and IL-1β mRNA expression, and apoptotic caspase 3 activity [74]. The underlying mechanisms by which quercetin exerts its beneficial effects were thought to be mediated through the suppression of changes in ERK/Akt phosphorylation and protein phosphatase activities in hippocampal slice and neuron/glia cultures [74]. Therefore, clinical studies should be undertaken to examine whether *Thymus atlanticus* and its compounds could offer an alternative tool for the prevention and treatment of stroke.

#### 4.2.6. Insulin Resistance

Previous investigations have shown that supplementation with a polyphenol extract of *Thymus Atlanticus* prevented the development of hyperglycemia and hyperinsulinemia in 63-day-old high-fat-fed hamsters [44]. The mechanism underlying these effects was thought to be attributed to the modulation of oxidative stress, as reflected by the reduction in the increase of malondialdehyde [44]. Ramchoun et al. [43] have reported that an aqueous extract and polyphenol fraction derived from *Thymus atlanticus* leaves had no effects on blood glucose levels but effectively prevented hyperinsulinemia in Triton-induced hyperlipidemia hamsters. The failure to obtain a significant decrease in blood glucose levels could be explained, at least in part, by the short duration of treatment with *Thymus atlanticus,* since 24 h was not enough to exert its beneficial effects on hyperinsulinemia as well as on hyperglycemia. Interestingly, studies have demonstrated that rosmarinic acid, one of the main polyphenols found in *Thymus atlanticus* [42], resulted in a decrease in insulin resistance index (HOMA-IR), a rise in the concentration of reduced glutathione, and a decrease in the concentration of advanced oxidation protein products in ovariectomized rats [75]. Moreover, rosmarinic acid was found to increase glucose uptake and inhibit insulin resistance in skeletal muscle cells through the activation of AMP-activated protein kinase (AMPK) [76,77]. Moreover, quercetin was reported to improve oral glucose tolerance as well as pancreatic β-cell function to secrete insulin and inhibit the α-glucosidase and DPP-IV enzymes, which prolong the half-life of glucagon-like peptide-1 [78]. Moreover, studies have shown that quercetin ameliorates insulin resistance and decreases oxidative stress in high-fat-fed mice [79]. The same authors have shown that quercetin promoted glucose uptake, repressed oxidative stress, and improved insulin resistance via regulating METTL3-mediated N6-methyladenosine modification of PRKD2 mRNA in skeletal muscle and C2C12 myocyte cell lines [79]. Rutin was found to increase glucose uptake through an elevated GLUT4 translocation induced by enhanced insulin receptor kinase activity in differentiated myotubes [80]. Thus, one may suggest that *Thymus atlanticus* and its compounds exert their beneficial effects on insulin action by decreasing oxidative stress.

## 5. Conclusions and Perspectives

*Thymus atlanticus* (Lamiaceae) is widely present in Morocco and constitutes an important source of bioactive molecules, which can be used for the treatment of infectious and non-infectious diseases, especially age-related diseases. The ease of cultivating *Thymus atlanticus*, which produces many bioactive molecules with significant therapeutic capacities, makes it possible to envisage a valuation that could, like argan oil, which is emblematic of Morocco, have positive economic repercussions at several levels.

The future direction must emphasize the characterization of novel bioactive compounds in *Thymus atlanticus* extracts as well as mixtures of compounds from *Thymus atlanticus* that may have therapeutic potential, such as antimicrobial, anti-inflammatory, or antioxidant properties. Moreover, safety and toxicology should be assessed, especially when used as dietary supplements, in phytomedicine, or in traditional medicine, to determine eventual side effects and establish safe dosage levels. Finally, yet importantly, to help conduct clinical trials on new *Thymus atlanticus* metabolites in well-designed clinical trials for specific health conditions and metabolomics, the latter may entail a deeper understanding of the metabolic changes induced by *Thymus atlanticus* extracts in living organisms.

## Figures and Tables

**Figure 1 nutrients-15-04077-f001:**
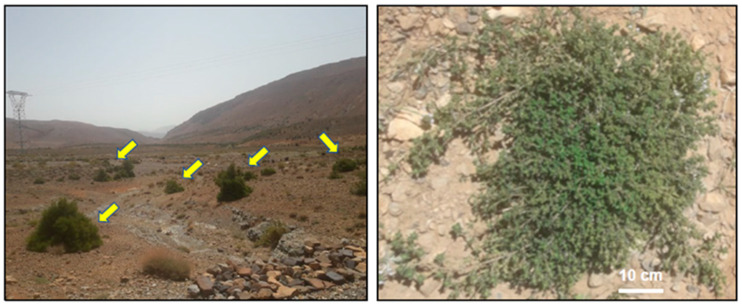
Photos of *Thymus atlanticus (Lamiaceae)*, High Atlas Mountain, Midelt Region, 2200 m (Morocco); yellow arrows point towards *Thymus atlanticus*.

**Figure 2 nutrients-15-04077-f002:**
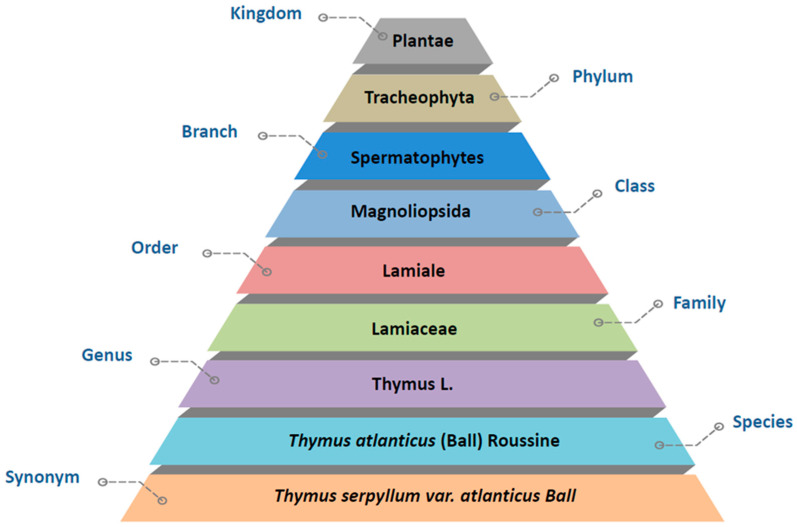
Botanical classification of *Thymus atlanticus*.

**Figure 3 nutrients-15-04077-f003:**
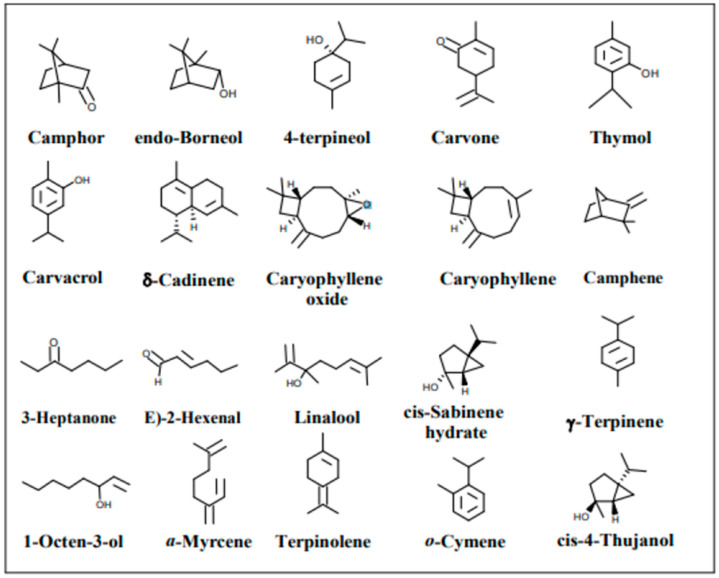
Structures of *Thymus atlanticus* essential oil components [30].

**Figure 4 nutrients-15-04077-f004:**
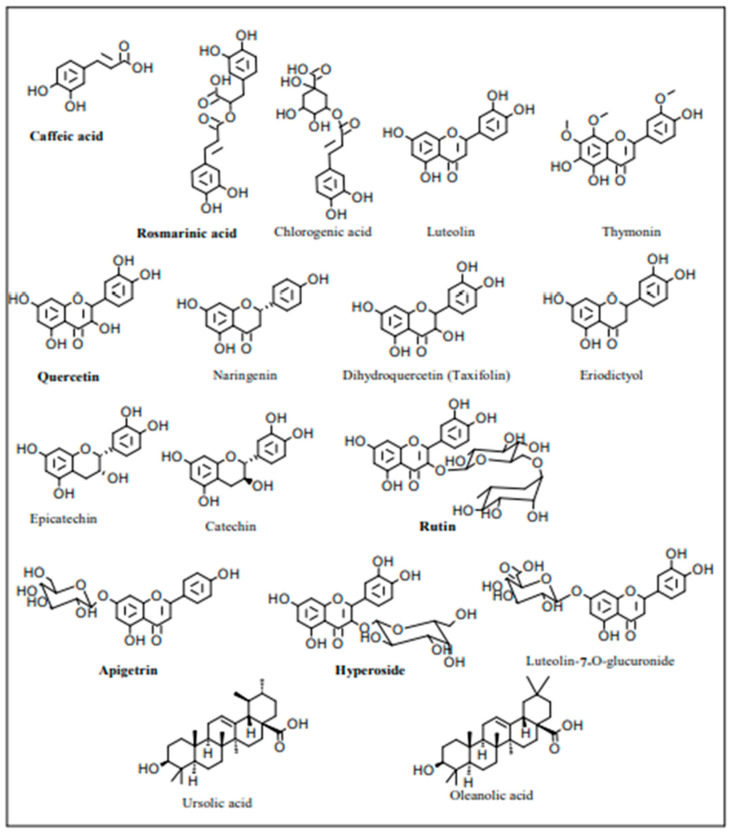
Structures of non-volatile secondary metabolites of the Thymus genus, including those of *Thymus atlanticus* species, are in bold [32].

## Data Availability

The datasets generated and analyzed for Appendix A are available from Dr Farid Khallouki, upon request.

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
