# Peer review of "Thymus atlanticus: A Source of Nutrients with Numerous Health Benefits and Important Therapeutic Potential for Age-Related Diseases"

_nutrients, 2023, doi:10.3390/nu15184077_

Round 1

Reviewer 1 Report

The submitted paper of the group of Midaoui et al. is an interesting review on Thymus atlanticus which deserves publication in Nutrients according to my point of view.

In principal my comment are of minor character.

The following issue I see as the most important. These two points should be improved.

1.       Some polishing of the paper as some information are uselessly repeated – see also below some examples in minor comments

2.       Lack of a chapter emphasizing that kinetic aspects, e.g. authors are frequently speaking about different flavonoids and their effects but they do not comment that these compounds have very low biovailability – examples are naringenin, apigenin at page 9, rosmarinic acid, quecetin, rutin at page 10 etc. (e.g. Systematic analysis of the polyphenol metabolome using the Phenol-Explorer database in Mol Nutr Food Res 2016 - doi: 10.1002/mnfr.201500435 or A Comprehensive Review of Rosmarinic Acid: From Phytochemistry to Pharmacology and Its New Insight. Molecules 2022 - doi: 10.3390/molecules27103292). Hence the observed effect reported e.g. in the chapter 4.2.4 can be due to small metabolites formed as was also recently suggested in this journal (3-Hydroxyphenylacetic Acid: A Blood Pressure-Reducing Flavonoid Metabolite – Nutrients 2022 - doi: 10.3390/nu14020328). Hence I strongly suggest to add a novel chapter 5 entitled e.g. Kinetic aspects and the possible effect of metabolites on the observed biological effects of Thymus atlanticus. It would be fine to discuss also the bioavailability of major components of its essential oil.

Other comments are rather of minor importance:

The paper has 15 authors, which is too much for a review paper based solely on 80 references, justification of all authors have to be hence specified in more detail in Author contributions, e.g. what was the role of each author – which chapters they were responsible for

Abstract -  It is also used for the realization of essential oil – isolation of essential oil

Page 2 - it was recommended for lung diseases and to avoid poisoning - ? – should be rewritten

Page 2 - Thyme can complement the treatment of many illnesses, including respiratory health – the expression should be more professional

Page 2 - Common thyme (Thymus vulgaris) has all the time been considered the best. Then comes the Spanish variety (Thymus Zygis).- Common thyme (Thymus vulgaris) has been considered the most suitable?/usable?/prestigious?/with the highest content of the active substances? .... There is also a Spanies variety (Thymus zygis) known also as white thyme.

Page 2 - neglected species, very locally abundant, may be threatened by excessive harvesting is the endemic Thymus atlanticus – I recommend rewriting of this sentence

Page 4 - The leaves 4–10 mm long - The leaves are 4–10 mm long

Page 5 - Some sparse data in chemistry of the Thymus genus – I do not agree, that the data are sparse

Page 5 – remove „(Spanish thyme, white thyme)“

Page 5- Essential Oils – essential oils

Page 6  - and Infraspecific variability [22,23]; Amarti et al [24]; Stahl-Biskup & Saez [4]. -  ?

Page 7 - As indicated in Figure 4, there are other polyphenolic compounds including apigenin, luteolin, and naringenin, thymonin, dihydroquercetin, eriodictyol, quercetin, naringenin, apigenin, catechin, rutin, apigetrin, caffeic acid, p-coumaric acid, chlorogenic acid and catechin – 2 x catechin

Page 8  - The non-volatile metabolites in Thymus atlanticus were described as rosmarinic acid; rutin; hyperoside; quercetin; apigetrin; caffeic acid – this is repetition of the sentence from the previous page – remove

Figure 5 – this is useless for this article, remove it please

Page 9 - Moreover, Thymus flavonoids such as apigenin and luteolin were reported to exert promising effects in overcoming antibiotic resistance – this is a too strong expression, I do not see clear data to support this hypothesis

Page 10 – Myocardial infarction – myocardial

Page 10 – hypertension - The mechanisms underlying the anti-hypertensive effects of rosmarinic acid were found to be mediated through its antioxidant properties [64,66], ACE inhibition [60,67], and vasodilation [64]. Discuss if such effects are really achievable in clinically obtainable concentrations / dose.

Page 10 - Animal studies have suggested that rosmarinic acid exerted neuroprotective effect against ischemic stroke through its anti-oxidative properties which involves upregulation of Nrf2 (Nuclear factor (erythroid-derived 2)-like 2) – this is incorrect expression, Nrf2 is activated upon oxidative stress so this is not an antioxidant effect but rather a consequence of a prooxidant effect

Page 11 - Extract and Polyphenol Fraction Derived – why upper case letters?

Page 11 - rosmarinic acid, the principal polyphenol found in Thymus Atlanticus – is it really true?, that rosmarinic acid is the principal polyphenol, this information should not be located here but in chapter 3.

Page 11 - Rutin, another Thymus atlanticus compound, - remove „another Thymus atlanticus compound“ – this was already mentioned in chapter

Page 11 - have rapid economic repercussions – more professional expression is neede

Check references 9, 10 and 12

fine, minor modification needed - see my comments

Author Response

Response to Reviewers

Manuscript Ref. No.: nutrients-2619339

Title: Thymus atlanticus: a source of nutrients with numerous health benefits and important therapeutic potential for age-related diseases.

Reviewer 1

 Lack of a chapter emphasizing that kinetic aspects, e.g. authors are frequently speaking about different flavonoids and their effects but they do not comment that these compounds have very low biovailability – examples are naringenin, apigenin at page 9, rosmarinic acid, quecetin, rutin at page 10 etc. (e.g. Systematic analysis of the polyphenol metabolome using the Phenol-Explorer database in Mol Nutr Food Res 2016 - doi: 10.1002/mnfr.201500435 or A Comprehensive Review of Rosmarinic Acid: From Phytochemistry to Pharmacology and Its New Insight. Molecules 2022 - doi: 10.3390/molecules27103292). Hence the observed effect reported e.g. in the chapter 4.2.4 can be due to small metabolites formed as was also recently suggested in this journal (3-Hydroxyphenylacetic Acid: A Blood Pressure-Reducing Flavonoid Metabolite – Nutrients 2022 - doi: 10.3390/nu14020328). Hence I strongly suggest to add a novel chapter 5 entitled e.g. Kinetic aspects and the possible effect of metabolites on the observed biological effects of Thymus atlanticus. It would be fine to discuss also the bioavailability of major components of its essential oil.

We thank the reviewer for his comments, which seem very relevant. However, it is noteworthy that the objective of our review is based on the scientific study of a Moroccan species of thyme on age-related diseases by highlighting the effects of Thymus atlanticus and its components on the development of age-related diseases with the elucidation of their potential involved mechanisms. We think that adding the Kinetic aspects and the possible effect of metabolites with the notion of biodisponibility will take us away from the topic.

The paper has 15 authors, which is too much for a review paper based solely on 80 references, justification of all authors have to be hence specified in more detail in Author contributions, e.g. what was the role of each author – which chapters they were responsible for.

As requested by the reviewer, all authors participation details are now included in author contributions of the revised manuscript.

Abstract -  It is also used for the realization of essential oil – isolation of essential oil

As requested by the reviewer, the word “realization” was replaced by “isolation”.

Page 2 - it was recommended for lung diseases and to avoid poisoning - ? – should be rewritten

As requested, the phrase was rewritten in the revised manuscript.

Page 2 - Thyme can complement the treatment of many illnesses, including respiratory health – the expression should be more professional.

As requested, the phrase was reformulated in the revised manuscript.

Page 2 - Common thyme (Thymus vulgaris) has all the time been considered the best. Then comes the Spanish variety (Thymus Zygis).- Common thyme (Thymus vulgaris) has been considered the most usable. There is also a Spanish variety (Thymus zygis) known also as white thyme.

The reviewer is thanked for his comment. The two phrases are now reformulated in the revised manuscript.

Page 2 - neglected species, very locally abundant, may be threatened by excessive harvesting is the endemic Thymus atlanticus – I recommend rewriting of this sentence

As requested, the sentence was reformulated in the revised manuscript.

Page 4 - The leaves 4–10 mm long - The leaves are 4–10 mm long

It was corrected.

Page 5 - Some sparse data in chemistry of the Thymus genus – I do not agree, that the data are sparse

The word “sparse” was removed.

Page 5 – remove „(Spanish thyme, white thyme)“

These words between brackets were deleted.

Page 5- Essential Oils – essential oils

They were replaced.

Page 6  - and Infraspecific variability [22,23]; Amarti et al [24]; Stahl-Biskup & Saez [4]. -  ?

The reference [4] corresponds to Stahl-Biskup & Saez, which was already mentioned in page 2.

Page 7 - As indicated in Figure 4, there are other polyphenolic compounds including apigenin, luteolin, and naringenin, thymonin, dihydroquercetin, eriodictyol, quercetin, naringenin, apigenin, catechin, rutin, apigetrin, caffeic acid, p-coumaric acid, chlorogenic acid and catechin – 2 x catechin

The reviewer is thanked for his comment. The second “catechin” word was deleted in the revised manuscript.

Page 8  - The non-volatile metabolites in Thymus atlanticus were described as rosmarinic acid; rutin; hyperoside; quercetin; apigetrin; caffeic acid – this is repetition of the sentence from the previous page – remove

The reviewer was right in pointing out that these molecules were already mentioned in page 7. However, the study of Kouya et al [34] is another investigation, which demonstrated the presence of these non-volatile metabolites in aqueous extract Thymus atlanticus. This information is now included in the revised manuscript.

Figure 5 – this is useless for this article, remove it please

As suggested by the reviewer, figure 5 is now removed in the revised manuscript. However, it was added as supplementary file.

Page 9 - Moreover, Thymus flavonoids such as apigenin and luteolin were reported to exert promising effects in overcoming antibiotic resistance – this is a too strong expression, I do not see clear data to support this hypothesis

The reviewer is thanked for his comment. The phrase was reformulated in the revised manuscript as following: “Moreover, Thymus flavonoids such as apigenin and luteolin were reported to exert partially positive effects against antibiotic resistance”.

Page 10 – Myocardial infarction – myocardial

The word "Myocardial" was replaced by "myocardial".

Page 10 – hypertension - The mechanisms underlying the anti-hypertensive effects of rosmarinic acid were found to be mediated through its antioxidant properties [64,66], ACE inhibition [60,67], and vasodilation [64]. Discuss if such effects are really achievable in clinically obtainable concentrations / dose.

The reviewer is thanked for its pertinent comment. Although rosmarinic acid seems to exert beneficial effects on experimental hypertension, human studies should be undertaken in order to confirm if such positive effects could be achievable in clinically obtainable concentrations per dose. This information is now included in the revised manuscript.

Page 10 - Animal studies have suggested that rosmarinic acid exerted neuroprotective effect against ischemic stroke through its anti-oxidative properties which involves upregulation of Nrf2 (Nuclear factor (erythroid-derived 2)-like 2) – this is incorrect expression, Nrf2 is activated upon oxidative stress so this is not an antioxidant effect but rather a consequence of a prooxidant effect

Animal studies have suggested that rosmarinic acid exerted neuroprotective effect against ischemic stroke through a mechanism involving the modulation of the Nrf2/HO-1 pathway. This information is now included in the revised manuscript.

Page 11 - Extract and Polyphenol Fraction Derived – why upper case letters?

The correction was done in the revised manuscript.

Page 11 - rosmarinic acid, the principal polyphenol found in Thymus Atlanticus – is it really true?, that rosmarinic acid is the principal polyphenol, this information should not be located here but in chapter 3.

This statement was modified in the revised manuscript as following: rosmarinic acid, one of the main polyphenol found in Thymus atlanticus.

Page 11 - Rutin, another Thymus atlanticus compound, - remove „another Thymus atlanticus compound“ – this was already mentioned in chapter.

As requested, it was removed in the revised manuscript.

Page 11 - have rapid economic repercussions – more professional expression is neede

The expression was modified in the revised manuscript.

Check references 9, 10 and 12

The references 9, 10 and 12 were checked and corrected.

Reviewer 2 Report

The authors collected wide range of information about Thymus atlanticus, but it is not structured well. There are multiple shortcomings in the MS. I have mentioned some points below-

1.      Write the name of all the plant in Italics throughout the MS.

2.      Write the full authentic name of the plant Thymus atlanticus along with its family.

3.      Define the source and timeline of the review from which information has been collected.

4.      Also include the traditional use of the Thymus atlanticus? Is there any information recorded in ancient or mythological texts? Consider and incorporate it.

5.      Draw all the structures in chemdraw or chemsketch and do not paste it in image form.

6.      The introduction should provide the relevant background and leads to well-defined objectives.

7.      What saponins, flavonoids, tannins, and alkaloids are present in Thymus atlanticus ? Explain.

8.      In Figure 5, define the retention time of rosmarinic acid and also add the UV spectra of rosmarinic acid from sample which can define the real specificity of compounds.

9.      Is there any quality control method reported for Thymus atlanticus ? If yes, please describe them.

10.  Overall we expect a critical assessment of the state of the art including precise and critical assessment of the papers reviewed (incl. concepts and methods).

11.  Conclusions need to be critical and specific. It needs to highlight the achievements and specific scientific gaps in our knowledge. So what further research should have priority?

12.  Make the references uniform.

There are moderate English language correction is required.

Author Response

Response to Reviewers

Manuscript Ref. No.: nutrients-2619339

Title: Thymus atlanticus: a source of nutrients with numerous health benefits and important therapeutic potential for age-related diseases.

Reviewer 2

  1. Write the name of all the plant in Italics throughout the MS.

The reviewer is thanked for his comment. As requested, the names of all the plants are now in Italics in the revised manuscript.

  1. Write the full authentic name of the plant Thymus atlanticusalong with its family.

The full authentic name of the plant Thymus atlanticus along with its family is now included in the revised manuscript.

  1. Define the source and timeline of the review from which information has been collected.

All references were checked.

  1. Also include the traditional use of the Thymus atlanticus? Is there any information recorded in ancient or mythological texts? Consider and incorporate it.

Following our exhaustive research, all information about Thymus in ancient or mythological texts was already incorporated in chapter 1.

  1. Draw all the structures in chemdraw or chemsketch and do not paste it in image form.

As requested, original Figures 3 and 4 are attached with the revised manuscript.

  1. The introduction should provide the relevant background and leads to well-defined objectives.

As requested by the reviewer, the introduction was modified in the revised manuscript.

  1. What saponins, flavonoids, tannins, and alkaloids are present in Thymus atlanticus ? Explain.

The objective was to review chemicals and biological properties of thymus atlanticus. This species is not yet well described scientifically since only a few very well known polyphenols have been described. Saponins, alkaloids, tannins, on the other hand, are still not known in the literature, there are only classical tests, which describe the presence of some of these metabolites but no identification yet.

  1. In Figure 5, define the retention time of rosmarinic acid and also add the UV spectra of rosmarinic acid from sample which can define the real specificity of compounds.

The reviewer is thanked for his comment. As suggested by the first reviewer, figure 5 is now removed in the revised manuscript. However, it was added as supplementary file.

  1. Is there any quality control method reported for Thymus atlanticus? If yes, please describe them.

No standardizations have been made yet.

  1. Overall we expect a critical assessment of the state of the art including precise and critical assessment of the papers reviewed (incl. concepts and methods).

We agree with this comment. In this context, additional information have been added in the paragraph 4.2 to support the benefits of thymus atlanticus in various diseases (infectious diseases and age-related diseases). It is now clearly written in this paragraph of the revised manuscript.

  1. Conclusions need to be critical and specific. It needs to highlight the achievements and specific scientific gaps in our knowledge. So what further research should have priority?

Changes in conclusions and perspectives section were made in the revised manuscript.

  1. Make the references uniform.

The references are now uniform in the revised manuscript.

There are moderate English language correction is required.

We performed English language corrections in the revised manuscript.
